# uPAR (PLAUR) Marks Two Intra-Tumoral Subtypes of Glioblastoma: Insights from Single-Cell RNA Sequencing

**DOI:** 10.3390/ijms25041998

**Published:** 2024-02-07

**Authors:** Yue He, Kristina B. V. Døssing, Maria Rossing, Frederik Otzen Bagger, Andreas Kjaer

**Affiliations:** 1Department of Clinical Physiology, Nuclear Medicine and PET & Cluster for Molecular Imaging, Copenhagen University Hospital—Rigshospitalet, 2200 Copenhagen, Denmark; yue.he@sund.ku.dk (Y.H.); kbdoessing@sund.ku.dk (K.B.V.D.); 2Department of Biomedical Sciences, University of Copenhagen, 2200 Copenhagen, Denmark; 3Center for Genomic Medicine, Rigshospitalet, Copenhagen University Hospital, 2100 Copenhagen, Denmarkfrederik.otzen.bagger@regionh.dk (F.O.B.); 4Department of Clinical Medicine, University of Copenhagen, 2200 Copenhagen, Denmark

**Keywords:** uPAR, *PLAUR*, single-cell RNA sequencing, GBM cell invasiveness, *CD44*, *FN1*, ECM degradation, inflammatory microenvironment

## Abstract

Urokinase plasminogen activator receptor (uPAR) encoded by the *PLAUR* gene is known as a clinical marker for cell invasiveness in glioblastoma multiforme (GBM). It is additionally implicated in various processes, including angiogenesis and inflammation within the tumor microenvironment. However, there has not been a comprehensive study that depicts the overall functions and molecular cooperators of *PLAUR* with respect to intra-tumoral subtypes of GBM. Using single-cell RNA sequencing data from 37 GBM patients, we identified *PLAUR* as a marker gene for two distinct subtypes in GBM. One subtype is featured by inflammatory activities and the other subtype is marked by ECM remodeling processes. Using the whole-transcriptome data from single cells, we are able to uncover the molecular cooperators of *PLAUR* for both subtypes without presuming biological pathways. Two protein networks comprise the molecular context of *PLAUR*, with each of the two subtypes characterized by a different dominant network. We concluded that targeting *PLAUR* directly influences the mechanisms represented by these two protein networks, regardless of the subtype of the targeted cell.

## 1. Introduction

Glioblastoma multiforme (GBM) is notorious for its poor prognosis and low survival rate [1,2,3]. One of the most deadly features of GBM is the infiltration of tumor cells into the surrounding tissue. Although migration beyond the brain is rare [4,5], intracranial migration can cause irreversible damage to the functional brain cells and hamper the complete surgical removal of tumor cells, leading to future recurrence [4,6]. Urokinase plasminogen activator receptor (uPAR), encoded by the *PLAUR* gene, is highlighted as a diagnostic marker for cell invasion in GBM [4,7]. uPAR has been reported to assist cell migration and angiogenesis, participate in the proteolytic process of extracellular matrix (ECM), regulate cell adhesion [8], and cell–matrix communication [9]. uPAR mediates the inflammatory responses within tumors by not only activating immune cells, such as neutrophils and macrophages, but also promoting immune cell infiltration [10]. Preclinical studies have shown its potential in peptide-based imaging [11], targeted radionuclide therapy [12,13], surgery [14], photothermal therapy [15], etc. The clinical feasibility of uPAR has also been widely recognized [16,17], leading to its advancement into phase II clinical trials with uPAR targeting positron emission tomography (uPAR-PET) [18,19,20,21,22,23,24].

Previous studies have primarily focused on specific aspects of uPAR, utilizing targeted assays like western blotting [25], cell transfection [25,26], and targeted RNA sequen- cing [27]. These approaches are adept at identifying molecular interactions among a group of proteins. However, they may confine the understanding of its molecular interactions to selected biological processes and pose a risk of partial or biased interpretation towards the selected molecules. In recent years, the single-cell RNA sequencing (scRNA-seq) platform has identified the invasive subtype of GBM [28,29,30] with a whole-transcriptome profile. This high-throughput technique made it possible to explore all the major biological features and molecular context of a marker gene with respect to the intra-tumoral subtypes. Despite this advantage, uPAR has not been specifically investigated regarding its subtype attributes in GBM.

In this study, we attempt to utilize *PLAUR* in GBM as a case study to establish a framework for exploring a particular biomarker using scRNA-seq datasets, regarding its molecular context and the biological hallmarks of the represented intra-tumoral subtypes. Using the scRNA-seq datasets from 37 GBM patients, we discovered that *PLAUR* is a marker gene for two main subtypes that feature ECM remodeling and inflammatory processes. *PLAUR* was found to be connected with two distinct protein networks in both subtypes, one associated with neutrophil activities and the other with cell migration.

## 2. Results

### 2.1. *PLAUR* Shows High Relative Expression within Single Cells

All 37 samples possess *PLAUR*-expressing cells. The abundance ranges from 1.6% to 56.7% (Figure 1a), showing high between-sample variations. The percentage rank is over 50% for 73.4% of all the *PLAUR*-expressing cells (Figure 1b). In 75.7% of the total 37 samples, the median percentage rank of *PLAUR* is over 50%. This implies a generally higher expression of *PLAUR* compared to other expressed genes.

### 2.2. Differential Expression Analysis of High-*PLAUR* Cells Reveals Involvement in Angiogenesis and Immune Response/Activation

A differential expression (DE) analysis was conducted between the high and low *PLAUR*-expressing cells to obtain differentially expressed genes (DEGs) in high *PLAUR*-expressing cells. Most of the cells express zero values of *PLAUR*; accordingly, we decided the criteria for high *PLAUR* expression to be the top 75% in the range between zero and the highest *PLAUR* level across the sample (Figure 2). The criteria were as follows: logFC > 2, and FDR < 0.05 were applied to select the DEGs. The DEGs in the high *PLAUR* cells from each sample were used to conduct enrichment analysis. The DEGs in the high *PLAUR* cells from samples P4, N3, N5, N11, N12, N14, N15, N16, N17, N18, N20, N21, and N22 reveal prominent Gene Ontology (GO) terms such as immune response and leukocyte activation (Table 1). In samples N2 and N4, the DEGs within the high *PLAUR* cells are characterized by key GO terms such as blood vessel development and angiogenesis (Table 1). The rest of the samples did not provide significant enrichment terms.

### 2.3. *PLAUR* Is Recognized as an Intratumoral Subtype Marker across Various Samples

The cells from each sample were clustered using the Louvain algorithm, followed by marker gene identification using Scanpy [31]. Only the clusters with the highest proportion of *PLAUR*-expressing cells were considered for further analysis. The marker genes for each of these clusters were selected using the following criteria: logFC > 2 and *p*-value < 0.05. *PLAUR* passes these criteria for 10 clusters from the following samples: N2, N5, N6, N12, N14, N17, N20, N21, N22, and D2 (Figure 3a,b). The over-expression of *PLAUR* by the *PLAUR* cluster is explicitly shown by an example from sample N20 in Figure 3c. For the 10 *PLAUR*-featuring clusters, *PLAUR* ranks in the top 1.3%, 2.7%, 3.1%, 5.6%, 5.9%, 8.9%, 28.3%, 37.3%, and 56.8%, 75.4% in terms of the *p*-value among all the marker genes (Figure 3d).

### 2.4. *PLAUR* Is a Marker Gene for Two Distinct Intra-Tumoral Subtypes

The clusters in which *PLAUR* meets the marker gene criteria logFC > 2 and *p*-value < 0.05 were further analyzed using a gene enrichment analysis using g:Profiler [32]. Only the top 200 marker genes ranked by *p*-value were used for the enrichment. The clusters from samples D2 and N2 showed similar biological features, and the clusters from samples N14, N20, N21, N12, N17, N22, and N5 share another feature. We named them “ECM-interaction subtype” and “Inflammatory subtype”, respectively. The *PLAUR*-represented cluster from N6 was found to undergo cell cycle activities (Appendix A) and thus removed from further studies.

To concisely present the biological features of these two subtypes, the criteria of the marker genes were tightened to logFC > 2 and *p*-value < 0.01, and only the shared marker genes between all the clusters of each subtype were used for the overall enrichment analysis. There are in total 147 shared markers between the clusters of the ECM-interaction subtype, and 192 shared markers between the clusters of the Inflammatory subtype.

For the Inflammatory subtype, the acute inflammatory condition is indicated by the predominating GO:BP terms associated with immune responses: “leukocyte activation”, “cell activation”, and “immune response”. The KEGG pathway suggested a more specific condition: “Staphylococcus aureus infection” (Figure 4). The ECM-interaction subtype is depicted by “extracellular matrix organization”, “cell adhesion”, and “wound healing” (Figure 4). The KEGG pathway describes a typical PLAUR-mediated cell–ECM interaction with “Focal adhesion”, “Complement and coagulation cascades”, and “Proteoglycans in cancer (Figure 4). In addition, angiogenesis and cell migration were indicated by “vasculature development” (*p* = 3 $\times \;10^{-6}$), “blood vessel morphogenesis” (*p* = 1.7 $\times \;10^{-5}$), “angiogenesis” (*p* = 3.4 $\times \;10^{-5}$), and “cell migration” (*p* = 9.5 $\times \;10^{-4}$, shown in Appendix A). It is worth noting that some typical features of the ECM-interaction subtype appear in the Inflammatory subtype with a slightly lower significance, and vice versa for the Inflammatory subtype (Appendix A).

### 2.5. *PLAUR* Primarily Operates within Two Protein Networks in Both Subtypes

A protein network analysis was conducted to unveil the molecular context of *PLAUR* in both subtypes. By combining the unique markers of all the clusters for each subtype under the criteria logFC > 2 and *p*-value < 0.01, we obtained 1870 unique marker genes for the Inflammatory subtype and 1792 for the ECM-interaction subtype. These marker genes were imported into Cytoscape using STRING App [33], and only the first neighbors of *PLAUR* were selected for further analysis. geneMANIA force-directed layout [34] was applied to visualize the different networks connected with *PLAUR*.

For both the Inflammatory and the ECM-interaction subtype, two protein networks were found to connect with *PLAUR* and its ligand, *PLAU* (Figure 5). An enrichment analysis was conducted using the nodes from each of the networks for both subtypes (Figure 6). Two networks featuring “neutrophil degranulation” and “positive regulation of cell migration” are the main connections of *PLAUR* regardless of the subtype. They were named “Neutrophil-activation network” and “Cell-migration network”, respectively. Nevertheless, the dominant network (the one with more nodes) is different for the two subtypes. The Neutrophil-activation network dominates the Inflammatory subtype (Figure 5a), and the Cell-migration network dominates the ECM-interaction subtype (Figure 5b).

These results are consistent with the fact that “regulation of cell-cell adhesion” (*p* = 3 $\times \;10^{-16}$) and “cell migration” (*p* = 1 $\times \;10^{-13}$) appear in the less significant range of the enrichment for the Inflammatory subtype, as does “neutrophil degranulation” (*p* = 3 $\times \;10^{-6}$) for the ECM-interaction subtype (Appendix A).

## 3. Discussion

As a biomarker advances in clinical use, verifying its represented subtype and molecular context becomes increasingly vital for validating its feasibility and managing risks. ScRNA-seq offers complete, impartial whole-transcriptome data without the necessity for pre-selected molecules, as in targeted assays. However, scRNA-seq-based analysis normally defines a subtype with a large set of marker genes. In a clinical setup of molecular imaging or targeted therapies, it is rare to find applications using more than two biomarkers [35,36,37]. Using the high-dimensional data of scRNA-seq, refining the information of one specific biomarker with a tolerable noise-to-signal ratio is challenging, since the information that we are looking for always blends with irrelevant signals, either from real biological processes or from noise. The fact that *PLAUR* exhibits an above-average expression level in the majority of cells, as indicated by the percentage rank, extricates its true characteristics from noise. This study demonstrated the possibility of obtaining valuable insights regarding one particular biomarker from scRNA-seq data. These results can find solid validations from previous studies.

The DE analysis was to simulate a bulk sequencing that compares two groups based on the *PLAUR* expression level. Although some major biological traits of *PLAUR* were found, the limitation persists in linking these features to specific cell subtypes, which is crucial for precise clinical targeting. Yet, it serves effectively as a concise overview of the marker alone.

In agreement with the biological features of the Inflammatory subtype, *PLAUR* expression was shown to correlate with the elevated inflammatory condition in the tumor [38]. A higher level of *PLAUR* has been reported in tumor-associated macrophages (TAMs) and other stromal cells in tumor microenvironments [39]. *PLAUR*-expressing cancer cells also facilitate macrophages to infiltrate tumor mass [40], and macrophages can promote *PLAUR* expression in tumor cells in return [41]. The KEGG pathway enrichment of the Inflammatory subtype presents a specific complication: “Staphylococcus aureus infection”. In fact, intracranial Staphylococcus aureus infection has been reported in GBM patients, caused by bacteremia via venous thromboembolism (VTE) [42]. VTE is an often-reported complication for tumors [43,44]. As a well-known fibrinolysis gene, *PLAUR* plays an essential role in the pro-coagulation state and highly correlates with immune response checkpoint genes, probably by interacting with leukocyte infiltration through VTE [45].

Regarding the ECM-interaction subtype characteristics, *PLAUR* is known to take an active role in the proteolytic degradation of ECM [46], and promoting the angiogenesis process [47,48]. It also favors cell migration either indirectly by ECM degradation or directly inducing pro-migratory activities [49]. Its connections with integrins plays a major role in focal adhesion [38], as shown by the KEGG pathways of the ECM-interaction subtype.

Using a protein network analysis, we speculate that *PLAUR* activates immune cells and assists their migration/adhesion to the inflammatory sites, and promotes tumor cell migration, mainly through ECM remodeling. *PLAUR* promotes neutrophil activation and degranulation while expressed on neutrophils [50,51], and also assists neutrophil infiltration to the inflammatory sites while expressed on cancer cells [52]. The Neutrophil-activation network highlights multiple integrins as major connections with *PLAUR*. The direct influence of uPAR–integrin complexes in neutrophil degranulation is yet to be discovered [51]. The migration network features multiple crucial cooperators of *PLAUR* in tumor cell migration. Among all of these, *MMP2*, *MMP9*, *FN1*, *CD44*, *SERPINE1*, and *CAV1* are the most famous collaborators with *PLAUR* in the cell migration of multiple cancer types [53,54,55,56,57,58]. The protein network analysis revealed the correlations of *PLAUR* with two major groups of proteins, regardless of the cell subtype. Although different studies have shown the functions of *PLAUR* and its interaction with different molecules, we present a comprehensive result including the primary roles and cooperators of *PLAUR* in GBM.

While scRNA-seq analysis has been effective in elucidating biological features, translating these findings into practical applications poses a challenge, since real-world applications often target proteins rather than RNA due to discrepancies in RNA and protein expression levels [59].

Future studies could explore the possibility of a better treatment response if combined with treatment with a second marker identified within the protein networks associated with *PLAUR*. This holds the potential to target pro-oncogenic signaling pathways in several subtypes and in both a receptor-dependent and -independent way. Treating more targets at once could lead to better treatment outcomes. In addition, if any of the proteins in the *PLAUR*-associated protein networks identified here are already targeted with an existing chemotherapeutic agent, it would allow clinicians to utilize treatments already in use in combination with other first-line treatments of GBM. Moreover, we recommend that any biomarker undergoing preclinical testing or earlier stages of testing utilize the developed pipeline in this study to ensure the precise targeting of the intended cell subtype.

## 4. Materials and Methods

### 4.1. Data Source and Pre-Processing

The datasets used for this study are 4 samples (1091 cells) from Darmanis et al. (GSE84465) [29], 5 samples (875 cells) from Patel et al. (GSE57872) [60], and 28 samples (7930 cells) from Neftel et al. (GSE131928) [28]. These samples are denoted as D1-D4, P1-P5, and N1-N28, respectively. FastQC [61] was applied to conduct quality control on FASTQ files of D1-D4 and P1-P5. The reads were trimmed using Trimmomatic [62]. Original reads were aligned using STAR [63]. The reference genome is the nucleotide sequence of the GRCh38 primary genome assembly downloaded from GENECODE database [64]. Adaptive criteria from Scater [65] were used to delete cells with too few reads or detected genes, and cells with excessively high mitochondrial genes. In addition, the mitochondria genes starting with “MT-” in their names and the genes summing up to less than 100 across all the cells were removed. The pre-processed data were normalized and log-transformed using the library size normalization provided by Scater [66]. The data from Neftel et al. (GSE131928) [28] were preprocessed by the authors and is available as Transcripts Per Million (TPM).

### 4.2. Calculation of Percentage Rank and Abundance

Two parameters were defined to characterize the expression level of a gene. “Abundance” is defined as the percentage of cells in a sample that has non-zero reads of the gene; “Percentage-rank”, represents the percentage of genes that express lower values than the gene in the same cell. We adopted the rank-based method to integrate datasets from different studies [67,68]. The percentage rank of genes for each cell was obtained by ranking all the non-zero gene reads ascendingly and multiplying by 100%. The abundance of a certain gene was defined as the percentage of cells expressing non-zero values of the gene from each sample.

### 4.3. Differential Expression Analysis

Differential expression (DE) was conducted between high and low *PLAUR*-expressing cells for each sample using edgeR [69]. The analysis was done using normalized data. Likelihood ratio test was applied and all the *p*-values were adjusted using the Benjamini–Hochberg (BH) method.

### 4.4. Cell Clustering and Enrichment Analysis

buildSNNGraph function from Scran [70] was used to find the nearest gene neighbors. Louvain clustering method from igraph [71] was applied to identify gene clusters. Ten nearest neighbors were applied during clustering. The clusters with the highest proportion of *PLAUR*-expressing cells were selected for further identification of *PLAUR*-represented clusters. The function “scanpy.tl.rank_genes_groups” from Scanpy [31] was used to calculate marker gene parameters. Marker genes were briefly selected using the following criteria: log fold change (logFC) > 2 and *p*-value < 0.05. Enrichment analysis was conducted using g:Profiler [32], and confirmed using DAVID database [72]. The *p*-values used in this study are all Benjamini–Hochberg adjusted.

### 4.5. Protein Network Analysis

The protein networks were formed using STRING APP [33] in Cytoscape [73]. GeneMANIA force-directed layout was used to visualize the networks [34]. Both the color and size of each node in the networks show the “degree” (the number of connections to other nodes) [74]. The bigger the node size and the redder the color both indicate a higher degree of the node.

### 4.6. Data Visualization

The box plots, bar plots, violin plots, and scatter plots were made using ggplot2 [75]. The volcano plots were made using EnhancedVolcano [76]. The t-distributed stochastic neighbor embedding (TSNE) plot was made by Scater [66]. The code for the entire study is available on GitHub (https://github.com/Emma920/scRNAseq-study-of-uPAR-PLAUR.git (accessed on 4 July 2023)).

## 5. Conclusions

By using two independent variables to assess *PLAUR* expression, our findings indicate that, while the ratio of the *PLAUR*-expressing cells varies among patients, it consistently demonstrates a higher expression level among the spectrum of expressed genes across the majority of samples. We discovered a correlation between elevated *PLAUR* expression and tumor angiogenesis and inflammatory activities through a differential expression analysis. The subsequent cell clustering unveiled *PLAUR* as a distinctive marker gene for specific cell subtypes within GBM. Further enrichment analysis revealed two cell subtypes characterized by *PLAUR*: one demonstrating intricate interactions with ECM, and the other highlighting notable inflammatory features. Delving into the protein network analysis further elucidated the molecular context surrounding *PLAUR*, revealing its primary connections with proteins that regulate cell migration and neutrophil activation in both cell subtypes of GBM. We concluded that targeting *PLAUR* directly influences the processes regulated by these two protein networks, independent of the cell subtype.

## Figures and Tables

**Figure 1 ijms-25-01998-f001:**
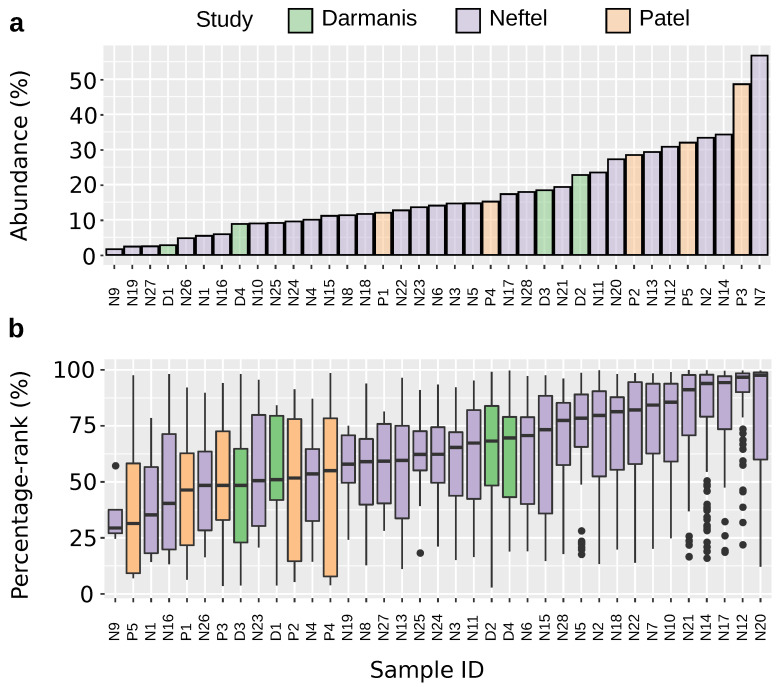
Expression level of *PLAUR* among patients. (**a**) The abundance of *PLAUR*-expressing cells across samples. (**b**) The percentage rank of *PLAUR* across samples. This calculation only includes cells that express *PLAUR*.

**Figure 2 ijms-25-01998-f002:**
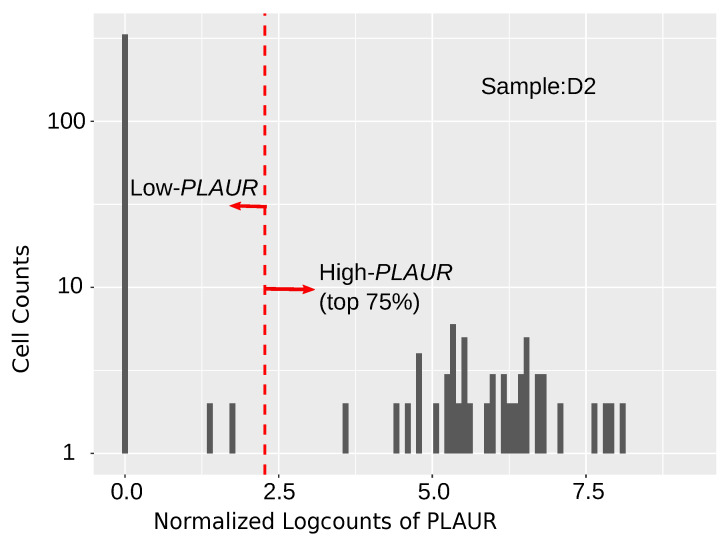
An example to illustrate the high *PLAUR* cells and low *PLAUR* cells for DE analysis in sample D2.

**Figure 3 ijms-25-01998-f003:**
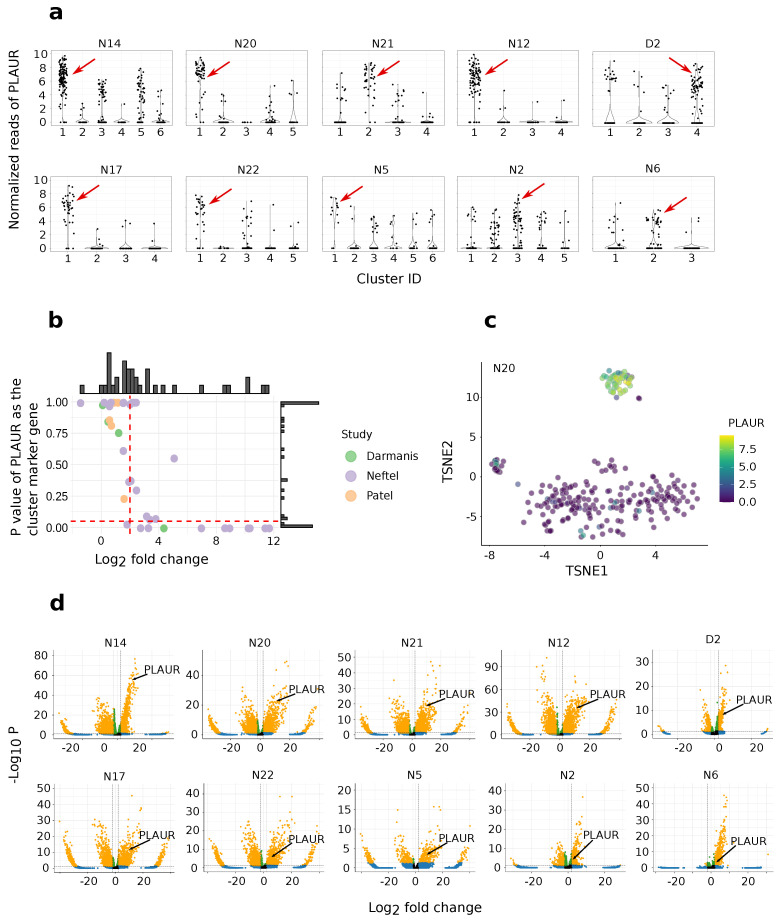
Discovering *PLAUR*-represented clusters. (**a**) Examples of *PLAUR* expression in respect to clusters. Each black dot represents a cell. The red arrows point to the clusters with the highest percentage of *PLAUR*-expressing cells. Note that the normalized reads are only comparable within the same sample. (**b**) A *PLAUR*-featuring cluster illustrated in TSNE from sample N20. (**c**) Log-fold changes versus the adjusted *p*-value of *PLAUR* as a cluster marker for the clusters that have the highest percentage of *PLAUR*-expressing cells in each sample. The horizontal red dash line is the adjusted *p*-value criteria *p* < 0.05 and the vertical red dashed line is logFC > 2. Selected clusters exhibiting *PLAUR* as their marker gene are in the right bottom corner. (**d**) Volcano plots for the clusters in which *PLAUR* passes the marker gene criteria. The criteria for the marker genes are logFC > 2 and *p*-value < 0.05. Markers that pass these criteria are colored in yellow. Over-expressed marker genes are on the right side of the coordinate.

**Figure 4 ijms-25-01998-f004:**
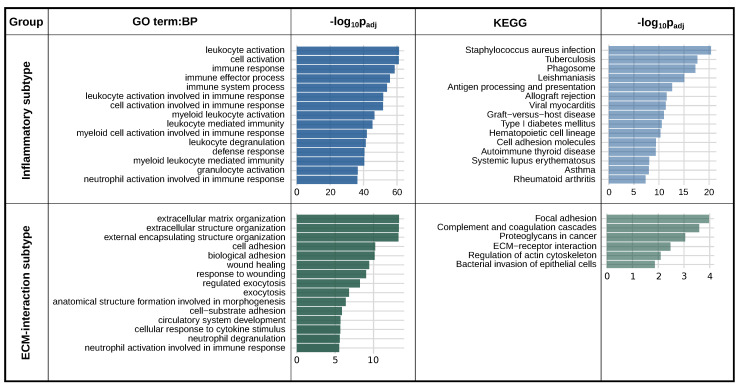
Enrichment results for the PLAUR-marked subtypes. GO:BP stands for GO terms: biological process. KEGG means KEGG pathways. The Inflammatory subtype results came from the shared marker genes across samples: N14, N20, N21, N12, N17, N22, and N5. The ECM-interaction subtype results are from samples D2 and N2. Only the top 15 terms ranked by *p*-value are shown. And there are only 6 significant KEGG terms for the ECM-interaction subtype. Only the top terms with sufficient significance (*p*-value < 0.05) are shown.

**Figure 5 ijms-25-01998-f005:**
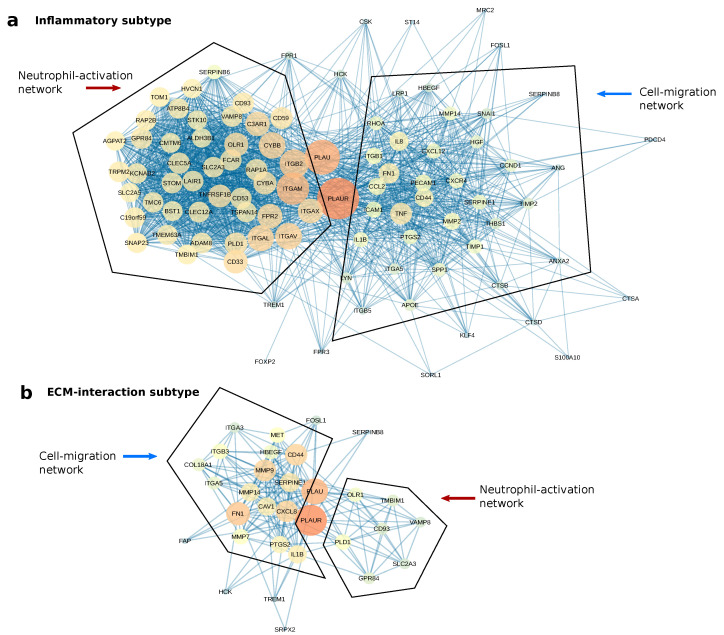
The protein networks of *PLAUR* for the two *PLAUR*-represented subtypes. Bigger and redder nodes have more connections (higher degree). (**a**) The two protein networks connected with *PLAUR* in the Inflammatory subtype. The networks were presented using the geneMANIA force-directed layout. (**b**) The two protein networks connected with *PLAUR* in the ECM-interaction subtype.

**Figure 6 ijms-25-01998-f006:**
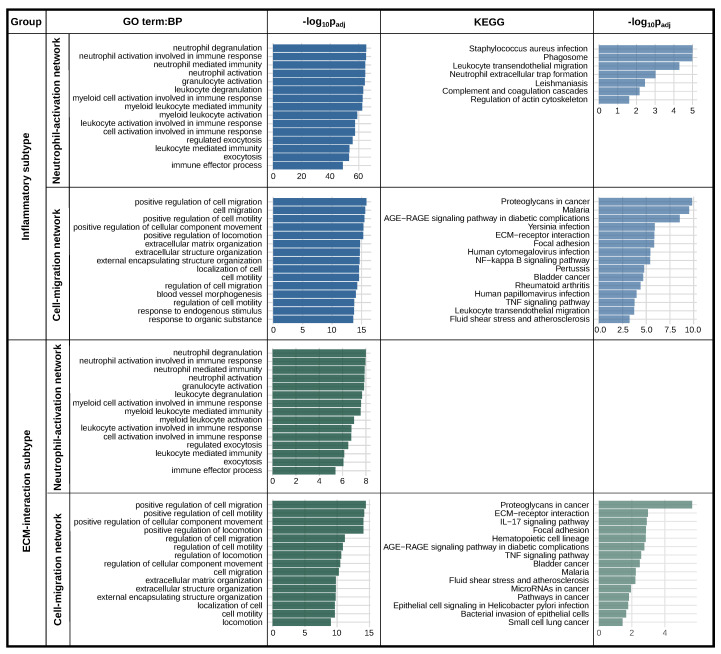
Enrichment analysis of the two protein networks for both of the *PLAUR*-represented subtypes. Only the top terms with sufficient significance (*p*-value < 0.05) are shown.

**Table 1 ijms-25-01998-t001:** Top 2 GO:BP enrichment results based on DEGs.

N2	N4	N3	N5	P4	N11	N12, N14, N15, N17, N18, N20, N21, N22	N16
response to external stimulus	angiogenesis	positive regulation of immune system process	immune response	immune effector process	leukocyte activation	immune system process	immune system process
blood vessel development	blood vessel morphogenesis	immune response	defense response	macrophage activation	cell activation	immune response	defense response

## Data Availability

Publicly available datasets were analyzed in this study. The data can be found here: https://www.ncbi.nlm.nih.gov/geo/ with accession numbers: GSE84465, GSE57872, GSE131928.

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
