# Peer review of "uPAR (PLAUR) Marks Two Intra-Tumoral Subtypes of Glioblastoma: Insights from Single-Cell RNA Sequencing"

_ijms, 2024, doi:10.3390/ijms25041998_

Round 1
Reviewer 1 Report
Comments and Suggestions for Authors
The manuscript investigates the expression patterns of PLAUR in the context of glioblastoma using only publicly available data sets. I have major concerns about the appropriateness of statistical modelling used and the lack of details of data preprocessing.
• The preprocessing steps are very much what I expect to see for bulk RNA-seq: "The reads were trimmed using trimmomatic. Original reads were aligned using STAR." How does this handle cell barcodes and UMIs? This doesn't seem appropriate to single-cell RNA data, unless the wording is missing key details. One key detail which is certainly missing is the reference genome used. hg19? hg38? CHM13?
• Library size normalisation from Scater was used to make different cells more similar. The developers of Scater caution that this method is simple but lacks biological realism, particularly for cancer data, which this manuscript is analysing.
"Strictly speaking, the use of library size factors assumes that there is no “imbalance” in the differentially expressed (DE) genes between any pair of cells. That is, any upregulation for a subset of genes is cancelled out by the same magnitude of downregulation in a different subset of genes."
Source: https://bioconductor.org/books/3.18/OSCA.basic/normalization.html#library-size-normalization
• Differential expression analysis between clusters of cells uses edgeR, which is a statistical method for small sample size bulk data, is not suited to differential expression between clusters of cells, with thousands of cells per cluster. The authors should refer to Confronting False Discoveries in Single-cell Differential Expression, Nature Communications, 2021. The single-cell counts matrix ought to be transformed into a pseudobulk matrix before hypothesis tests are performed.
• There is a lack of detail about the clustering process. The number and size of clusters depends on a user-customisable resolution parameter. What values of the resolution parameter did the authors evaluate and why was any particular value chosen? For a sound way of choosing the clustering resolution, the authors should refer to Significance Analysis for Clustering with Single-cell RNA-Sequencing Data, Nature Methods, 2023.
• Protein Network Analysis. This aspect of the analysis depends strongly on the STRING database and make a leap of faith by labelling genes highly expressed in the same clusters as PLAUR as having "molecular interactions" with PLAUR. This is the perennial correlation ≠ causation problem. To be convincing, I would expect to see affinity purification-mass spectrometry data for glioblastoma with PLAUR as the bait protein.
In summary, most of the analyses have insufficient detail and seem to be unsuitable for the kind of data which was collected. Also, the claims made are too far-fetched for the type of observational data which is presented in the manuscript.
Comments on the Quality of English LanguageTitle could be improved. Current wording made me think that there were two subtypes / isoforms of PLAUR identified in RNA sequencing, but it actually refers to the cancer subtypes. Perhaps something like: uPAR(PLAUR) Marks Two Intra-tumoral Subtypes of Glioblastoma: Insights from Single-cell RNA Sequencing.
Reviewer 2 Report
Comments and Suggestions for Authors
The manuscript is well designed and very nicely written. However, please address some comments below:
1. For the GO analysis, you used gprofiler. It would be worth to try also to analyze your data using another online GO tool for example: metascape, David Bioinformatics, etc. to compare and validate your data, and to draw final conclusions. 2. Similarly, for the protein-protein interactions that you analyzed using STRING, it would be great to validate the results using also another platform like Cytoscape and to compare the results. 3. A more detailed description of potential future implications of your findings and potential further research ideas should be discussed. 4. Please include limitations of your study.Author Response
Please see the attachment

Reviewer 3 Report
Comments and Suggestions for Authors The article “uPAR(PLAUR) features two distinct intra-tumoral subtypes in glioblastoma: Insights from single-cell RNA sequencing uPAR(PLAUR) features two distinct intra-tumoral subtypes in glioblastoma: Insights from single-cell RNA sequencing” is very interesting. I only have one question What could the KEGG pathway suggest other conditions such as Tuberculosis or Leishmaniasis? Thank you so muchAuthor Response
Please see the attachment.

Reviewer 4 Report
Comments and Suggestions for Authors
The authors explore the role of PLAUR, as a clinical marker in glioblastoma multiforme (GBM). They utilize available single-cell RNA sequencing data to identify PLAUR as a marker for two subtypes of GBM—one characterized by inflammatory activities and the other by extracellular matrix (ECM) remodeling. The study reveals molecular cooperators for both subtypes without assuming specific pathways, emphasizing two distinct protein networks associated with PLAUR. The manuscript can be accepted once the following questions have been addressed.
I would appreciate more information on the methodology if you integrate scRNA-seq data from different studies, such as, removing batch effects and normalizing the integrated dataset? It would be helpful if you could include visualizations, such as UMAP or t-SNE plots, to demonstrate the effectiveness of the integration process. Would you be able to determine whether the PLAUR high and low expressed cells are dispersed or concentrated in a specific region?
The authors utilized edgeR to identify differentially expressed genes in single cells. Another paper (DOI: 10.1007/978-1-4939-9240-9_25) demonstrated that edgeR, using the quasi-likelihood F-test (QLF), outperforms other methods in detecting differentially expressed genes between two groups of single cells. Could you please provide more details about the usage of edgeR in your analysis? Are the raw read counts or normalized read counts used as input here?
The authors defined low and high PLAUR cells based on various single-cell RNA sequencing (scRNA-seq) studies. They discovered that these cells exhibited two similar biological characteristics across multiple datasets, namely ECM (D2, N2) and inflammatory (other 7). PLAUR has already been defined as a biomarker for cell invasion in GBM. However, this type of analysis does not provide much novel evidence. Could you please tell what type of cells were annotated in previous studies for these high PLAUR cells? I am wondering if ECM and inflammatory cells represent two different types of cells.
Typo, line 91 “The volcano plots were made using ([49]).”
Round 2
Reviewer 4 Report
Comments and Suggestions for Authors
I thank the authors for addressing my concerns.